# Association between Biomarkers of Cardiovascular Diseases and the Blood Concentration of Carotenoids among the General Population without Apparent Illness

**DOI:** 10.3390/nu12082310

**Published:** 2020-07-31

**Authors:** Mai Matsumoto, Naoko Waki, Hiroyuki Suganuma, Ippei Takahashi, Sizuka Kurauchi, Kahori Sawada, Itoyo Tokuda, Mina Misawa, Masataka Ando, Ken Itoh, Kazushige Ihara, Shigeyuki Nakaji

**Affiliations:** 1Innovation Division, KAGOME CO. LTD., 17 Nishitomiyama, Nasushiobara, Tochigi 329-2762, Japan; Naoko_Waki@kagome.co.jp (N.W.); Hiroyuki_Suganuma@kagome.co.jp (H.S.); 2Graduate School of Medicine, Hirosaki University, 5 Zaifu-cho, Hirosaki, Aomori 036-8562, Japan; ippei@lyremizoguchi.com (I.T.); s_kurauchi@auhw.ac.jp (S.K.); iwane@hirosaki-u.ac.jp (K.S.); i-tokuda@hirosaki-u.ac.jp (I.T.); m_misawa@hirosaki-u.ac.jp (M.M.); masataka.ando@hirosaki-u.ac.jp (M.A.); itohk@hirosaki-u.ac.jp (K.I.); ihara@hirosaki-u.ac.jp (K.I.); nakaji@hirosaki-u.ac.jp (S.N.)

**Keywords:** carotenoid, vegetable intake, metabolic syndrome, healthy subjects, resident-based cross-sectional study

## Abstract

Several studies have demonstrated that carotenoid-rich vegetables are useful against cardiovascular diseases (CVDs). However, it is still unclear when a healthy population should start eating these vegetables to prevent CVDs. In this study, we evaluated the role of carotenoids in CVD markers in healthy subjects using age-stratified analysis. We selected 1350 subjects with no history of apparent illness who were undergoing health examinations. We then evaluated the relationship between the serum concentrations of six major carotenoids as well as their total, and nine CVD markers (i.e., body mass index (BMI), pulse wave velocity (PWV), systolic blood pressure (SBP), diastolic blood pressure (DBP), Homeostatic Model Assessment of Insulin Resistance (HOMA-IR), blood insulin, fasting blood glucose (FBG), triglycerides (TGs), and high-density lipoprotein (HDL) cholesterol) using multiple regression analysis. It was found that the total carotenoid level was significantly associated with seven markers other than BMI and FBG in males and with eight markers other than DBP in females. Many of these relationships were independent of lifestyle habits. Many significant relationships were found in young males (aged 20–39) and middle-aged females (aged 40–59). These findings can be used as lifestyle guidance for disease prevention although the causal relationships should be confirmed.

## 1. Introduction

The principal role of food, unlike medicine, is to prevent diseases by maintaining our bodies in a healthy condition over time. Many reports have suggested that higher intake of vegetables and fruits decreases the risk of chronic diseases, such as cancer and cardiovascular diseases (CVDs) [1,2,3,4,5,6]. In a systematic review, Aune et al. showed that the summary relative risk per 200 g of vegetables per day is 0.9 for CVDs [7]. Various components in vegetables have been reported to have such effect, with carotenoids being among the most prominent.

Carotenoids are red, orange, and yellow pigments found abundantly in vegetables and fruits, with more than 600 types identified in nature [8]. However, carotenoids commonly found in human blood are limited, and most of the carotenoids that we ingest in our normal diets belong to only six types (lutein, zeaxanthin, β-cryptoxanthin, α-carotene, β-carotene, and lycopene). The distribution of carotenoids in organs and tissues varies according to the carotenoid type [9,10]. In addition, each type of carotenoid has a specific expected physiological effect. For example, lutein, which is abundant in green leafy vegetables, might prevent age-related macular degeneration [11]. Ingesting tomato products containing large amounts of lycopene may be useful in preventing prostate cancer [12,13]. As stated earlier, most of the previous studies investigating the health benefits of carotenoids have focused primarily on the effects of specific carotenoids in patients with specific diseases.

Active consumption of various vegetables and fruits, rather than specific carotenoids, is generally recommended [14] for the prevention of chronic diseases. Various epidemiological studies suggest that higher intake of vegetables and fruits decreases the risk of CVDs by lowering the blood pressure, reducing the levels of proinflammatory cytokines and inflammatory markers, and improving insulin sensitivity [15], as noted earlier. As humans cannot synthesize carotenoids in their bodies, the concentration of accumulated carotenoids is considered to reflect the intake of vegetables and fruits. In fact, the total concentration of lutein, β-cryptoxanthin, α-carotene, β-carotene, and lycopene in the blood shows a stronger correlation with vegetable and fruit intake compared to the concentration of each carotenoid individually [16]. In addition, all of these carotenoids have a strong singlet oxygen elimination capability [17], suggesting that they may generally contribute to the improvement of CVDs. Thus, when evaluating blood carotenoid levels with the aim of preventing CVDs, total carotenoid concentrations need to be considered in addition to individual carotenoids. In addition, not only dietary habits, but also lifestyle, such as smoking, might affect the blood concentration of these carotenoids [18].

As mentioned above, many lines of evidence have suggested that carotenoids are effective in preventing CVDs. However, a large percentage of healthy people are unaware of the importance of prevention, and it may take triggers for them to change behaviors such as proactive vegetable intake. One effective method is to provide personalized information. For example, information on when to start eating carotenoid-rich vegetables would be useful. However, to the best of our knowledge, no research has so far provided such information. Therefore, as a first step to obtain such information, we conducted a cross-sectional study on the associations between serum concentrations of carotenoids and various health parameters associated with CVDs, including lifestyle, in a relatively healthy population with different ages.

## 2. Materials and Methods

### 2.1. Study Design and Subjects

This cross-sectional study was based on the Iwaki Health Promotion Project, an annual health examination conducted by Hirosaki University for the residents of the rural area of Hirosaki City, Aomori Prefecture, Japan, in May and June 2015–2018. Out of a total of 1759 participants, 1350 (538 males, 812 females) were included as healthy subjects. These subjects had complete clinical data, were not on any medication for dyslipidemia, and had no history of serious diseases, such as cancer, stroke, CVDs, kidney diseases, liver diseases, or diabetes. We used data from the earlier year for those who had multiple checkup visits. All procedures, as well as subject recruitment, were conducted in accordance with the Declaration of Helsinki and were approved by the ethics boards of Hirosaki University School of Medicine (2014-377, 2016-028, 2017-026, 2018-012, and 2018-063) and KAGOME CO., LTD (2015-R04, 2016-R03, 2017-R06, and 2018-R02), and all subjects provided written informed consent.

### 2.2. Self-Administered Questionnaire

A self-administered questionnaire was provided to the participants in advance of the health examination and was collected on the day of blood sampling. The questionnaire contained information about sex, age, current smoking habits, exercise habits, medical history, and use of medications. We defined subjects who engaged in exercise for more than 30 min per day at least twice a week habitually throughout the year as being accustomed to exercise. The volume of alcohol consumed (g/day) and that of vegetables consumed (g/day) were estimated using a brief-type self-administered diet history questionnaire (BDHQ) [19]. This BDHQ calculates the daily intake for key food groups on the basis of the past month’s dietary surveys. Notably, this questionnaire has been validated and is commonly used as a dietary survey in Japan [20].

### 2.3. Body Measurements

Body mass index (BMI) was calculated from the body height and weight, which were obtained by anthropometric measurements. Brachial ankle pulse wave velocity (baPWV) was measured using a volume plethysmograph (Form PWV/ABI; OMRONCOLIN Co Ltd., Tokyo, Japan). Systolic blood pressure (SBP) and diastolic blood pressure (DBP) were measured using a mercury manometer.

### 2.4. Blood Sampling and Testing

Blood was collected from the median cubital vein after overnight fasting. Measurements of blood biomarkers related to diabetes and dyslipidemia were consigned to LSI Medience Corporation (Tokyo, Japan). The concentrations of fasting blood glucose (FBG), triglycerides (TGs), and high-density lipoprotein (HDL) cholesterol were measured using enzyme assays, and the blood insulin concentration was determined using a chemiluminescent immunoassay. Homeostatic Model Assessment for Insulin Resistance (HOMA-IR) values were calculated using the FBG and blood insulin concentrations. The concentration of blood antioxidants was measured by KAGOME CO., LTD (Nagoya, Japan). Vitamin A (retinol), vitamin E (α-tocopherol), and carotenoids (lutein, zeaxanthin, β-cryptoxanthin, α-carotene, β-carotene, and lycopene) were extracted from the serum as previously described [21], and the serum concentrations were determined using a high-performance liquid chromatograph with a photodiode array detector [22]. The plasma concentration of vitamin C (ascorbic acid) was measured using a commercially available kit (R01K02; Shima Laboratories Co. Ltd., Tokyo, Japan).

### 2.5. Statistical Analyses

The mean value of each measurement was stratified by sex (male and female) and compared using the Mann–Whitney U test and was also stratified by age category (young, 20–39 years; middle-aged, 40–59 years; old ≥60 years) and compared using the Kruskal–Wallis test (post hoc: Bonferroni).

Correlation analysis of each carotenoid level was performed using Spearman’s rank correlation coefficient. Multiple regression analysis was performed, stratified by sex, to evaluate the relationships between the serum concentrations of carotenoids and vegetable intake. Serum concentrations of vitamins A and E and plasma concentration of vitamin C were also adopted as adjustment factors for the evaluation of total carotenoids.

Multiple regression analysis stratified by sex and age was also conducted to evaluate the relationships between the serum concentrations of carotenoids and nine biomarkers (BMI, baPWV, SBP, DBP, HOMA-IR, blood insulin, FBG, TGs, and HDL cholesterol), which are predictive biomarkers of obesity, arteriosclerosis, hypertension, diabetes, and dyslipidemia. These nine biomarkers and vegetable intake were adopted as objective variables, whereas carotenoids were adopted as explanatory variables. Age and use of antihypertensive medication were adopted as adjustment factors for all multi-regression analyses, and the volume of alcohol consumed, smoking habits, exercise habits, and BMI were also adopted as adjustment factors according to the analysis. Serum concentrations of vitamins A and E and plasma concentration of vitamin C were also adopted as adjustment factors for the evaluation of total carotenoids. All analyses were performed using the R statistical package (version 3.5.0, R Foundation for Statistical Computing, Vienna, Austria), and a *p*-value of <0.05 was considered statistically significant.

## 3. Results

### 3.1. Characteristics of the Study Subjects

Table 1 shows the mean values of all measurements stratified by sex and age categories, and many differences were found among sex and age groups. Most of the CVD markers worsened with age. However, two markers (BMI and TG) showed the highest value in middle-aged males and two markers (blood insulin and HDL cholesterol) began to worsen significantly in older females. Vegetable intake calculated from the dietary survey was found to be higher in females than in males. When compared by age, the average was lower in the young group and higher in the old group for both males and females. The serum concentrations of carotenoids behaved similarly to vegetable intake. In terms of differences, the level of zeaxanthin did not differ between males and females, whereas the level of lycopene was higher in younger than in older subjects.

### 3.2. Relationship between Serum Total Carotenoid Concentration and Vegetable Intake

Multiple regression analysis was performed using vegetable intake as an objective variable, blood total carotenoid concentration as an explanatory variable, and antioxidative vitamins (vitamin C (ascorbic acid), vitamin E (α-tocopherol), and vitamin A (retinol)), to which some carotenoids were internally converted, as adjustment factors. The results showed that the serum concentrations of total carotenoids showed a significant positive relationship with total vegetable intake in both males and females (male standardized partial regression coefficient (std. β) = 0.207, *p* < 0.001; female std. β = 0.201, *p* < 0.001).

### 3.3. Relationship between Serum Concentrations of Carotenoids and Markers of CVDs

Multiple regression analysis was performed using the nine abovementioned CVD markers as objective variables and the serum concentrations of total carotenoids as explanatory variables for the three age categories and in total for each sex. The standardized partial regression coefficients for each of these are shown in Table 2 and Table 3 and are compared in the presence and absence of adjustment factors.

When we applied age, antihypertensive drug intake, and vitamins as adjustment factors in the analysis, we found that the serum concentrations of total carotenoids were significantly associated with seven markers (baPWV, SBP, DBP, HOMA-IR, blood insulin, TGs, and HDL cholesterol) in males (Table 2) and with eight markers (BMI, baPWV, SBP, HOMA-IR, blood insulin, FBG, TGs, and HDL cholesterol) in females (Table 3). In males, the relationships between the serum concentrations of total carotenoids and glucose metabolism-related markers (HOMA-IR and blood insulin) disappeared by age stratification in all age groups. However, many relationships remained significant in females, with significant associations being lost in multiple parameters (baPWV, SBP, and FBG) only in the older group.

These relationships without age stratification remained even when adjustment factors of lifestyle were added (Pattern 2). In the age-stratified analysis, the serum total carotenoid concentrations associated significantly with all nine biomarkers in middle-aged females; these relationships were negative except for that with HDL cholesterol.

BMI was significantly and negatively related to the serum concentrations of total carotenoids in all age categories in females. Therefore, we added BMI as an adjustment factor to the analyses (Pattern 3). In females, as shown in Table 3, several relationships remained significant in the middle-aged group. However, many of the relationships observed in the younger and older females disappeared. In young males, a larger number of significant associations were found compared to other age groups.

### 3.4. Relationship between Individual Carotenoid Levels and Markers of CVDs

In the previous chapter, we discussed total carotenoids. As mentioned in the Introduction, different carotenoids may behave differently. Therefore, we also evaluated individual carotenoids. Multiple regression analysis was performed using CVD markers as objective variables and individual carotenoid concentrations as explanatory variables. We removed α-carotene as an explanatory variable because it showed a strong correlation with β-carotene (ρ = 0.836, *p* < 0.001). Other correlations between individual carotenoids are shown in Appendix A. Table 4 and Table 5 show all the standardized partial regression coefficients.

Not all carotenoids displayed the same behavior. For example, β-carotene behaved similarly to the total carotenoid concentration in both males and females. In females, serum concentration of lutein also showed a similar association to that of total carotenoids. In males, serum concentration of lutein was negatively associated with glucose-metabolism-related markers (HOMA-IR and insulin). Additionally, β-cryptoxanthin showed a similar association to that of total carotenoids in males. Lycopene was negatively associated with baPWV and positively associated with HDL cholesterol in both males and females. Only lycopene was not significantly associated with BMI in females. In addition, HDL cholesterol was positively associated with all carotenoids except for β-carotene in males. Overall, baPWV and blood pressure were associated with β-carotene and β-cryptoxanthin in both males and females, and glucose metabolism was associated with all carotenoids other than lycopene.

By adding adjustment factors of lifestyle in both males and females, the significant negative association between β-cryptoxanthin and TG disappeared (Pattern 2). On the other hand, in the analysis in which BMI was added as an adjustment factor, in females many of the associations between β-cryptoxanthin and CVD markers disappeared, and a positive relationship between serum zeaxanthin and DBP appeared (Pattern 3).

## 4. Discussion

In the present study, we analyzed the role of vegetable intake in the prevention of CVDs in a healthy Japanese population by clarifying the relationships between carotenoids, an indicator of vegetable intake and a natural strong antioxidant, and markers of CVDs. Several mechanisms of carotenoid-rich vegetables for the prevention of CVDs were inferred from previous reports (Figure 1).

Even in healthy individuals without apparent illness, we found statistically significant relationships between serum carotenoids and various CVD markers, and some of these relationships varied according to sex and age categories. To the best of our knowledge, this is the first study investigating the relationships between serum carotenoid levels and multiple CVD markers in healthy individuals by identifying differences in relation to sex and age.

### 4.1. Characteristics of the Study Subjects

As shown in Table 1, the levels of many biomarkers appeared to increase with age in our subjects. For example, the average baPWV in the old group seemed to be relatively high compared to the cutoff value (1750 cm/s) in a previous study [23]. The average baPWV in the Takashima Study (mean age: 58.9 years), based on an annual community health examination for the residents of Takashima City, Shiga Prefecture, Japan, was 1548 cm/s [24], similar to the average of our participants aged 40 or above (i.e., 1520 cm/s; mean age: 58.0 years). Thus, there is no reason to think that the population in this study is not a general, relatively healthy population. According to the guidelines for hypertension issued by the Japanese Society of Hypertension, the mean SBP in advanced-age males falls within the range of high normal blood pressure (130–139 mmHg) [25]; however, it does not reach the level of high blood pressure (140 mmHg or above) in Japanese subjects. In addition, BMI was higher in the middle-aged group for males compared to the old group, and it increased with age for females. Thus, the trend observed in this population was similar to that observed in the National Health and Nutrition Survey (Japan, 2017) [26]. In total, 17.1% of the males and 18.0% of the females were on antihypertensive drugs, values that are lower than the results of the National Health and Nutrition Survey (31.8% for males and 25.9% for females). This may be due to the fact that we focused on a healthier population in this study. The serum concentrations of carotenoids were consistent with previous reports [27]. These results suggest that the study population is a relatively general and healthy Japanese population.

Serum total carotenoid levels were significantly positively associated with vegetable intake in both males and females. In a recent systematic review, it was suggested that the concentrations of some carotenoids (lutein, β-cryptoxanthin, α-carotene, and β-carotene) are higher in the high-intake group of fruits and vegetables than in the low-intake group [28]. A relationship between blood carotenoid levels and vegetable intake was also reported by Campbell et al. using a frequency survey Food Frequency Questionnaire [16]. These previous studies are consistent with ours, suggesting that serum carotenoid concentrations could be an indicator of vegetable intake and that this could be expected to be helpful for nutritional guidance according to which non-invasive measurement of skin carotenoid levels has begun to be implemented [29].

### 4.2. Relationship between Serum Total Carotenoid Levels and Markers of CVDs

Without age stratification, in males, seven CVD markers (baPWV, SBP, DBP, HOMA-IR, insulin, TGs, and HDL cholesterol) out of the nine were significantly associated with total carotenoids (Table 2), whereas in females, eight markers (BMI, baPWV, SBP, HOMA-IR, blood insulin, FBG, TGs, and HDL cholesterol) were associated with total carotenoids (Table 3). Many studies have shown that some CVD markers are healthier in those with higher vegetable intake [4,30,31]. In addition, it has been suggested that the relationship between vegetable intake and some CVD markers is due to antioxidants such as carotenoids in the blood [32,33]. These results are consistent with ours.

The best studies of physiological function of carotenoids have focused on quenching the activity of singlet oxygen, one of the reactive oxygen species (ROS) [34]. ROS have been considered to elicit chronic inflammation and accelerate the onset and/or progression of CVDs [35]. In the present study, we did not evaluate any circulating biomarkers of early endothelial dysfunction, which is the starting point of the atherosclerotic process and accompanies the progression of CVDs [36]. We suggested that tomatoes can help prevent atherosclerosis by inhibiting vascular endothelial dysfunction in a primitive animal experiment [37]. Tsitsimpikou et al. suggested that continuous intake of tomato juice significantly improves inflammation status and endothelial dysfunction [38]. Regarding these mechanisms, reduction of the total expression of VCAM-1 and ICAM-1 by carotenoids (β-carotene and lycopene) has been reported to play an important role. [39].In the future, we aim to elucidate the mechanism in detail by examining the relationship between internal carotenoid levels and markers such as E-selectin, ICAM-1, VCAM-1, and von Willebrand Factor as well as inflammatory markers. We found significant associations between hydrocarbon carotenoids (β-carotene, lycopene) and the level of baPWV. Arteriosclerosis progresses via the deposition of lipid in blood vessel walls, which is induced by foamy macrophages that infiltrate vascular endothelial cells and absorb oxidized LDLs without restriction [40]. It has been reported that carotenes, which are mainly distributed in LDL cholesterol particles in the blood, improve the oxidative resistance of LDLs and reduce the levels of oxidized LDLs [41,42]. They are also expected to have an inhibitory effect on the progression of arteriosclerosis and may contribute to the negative relationships observed between carotenoids and baPWV. Oxidative stress has also been reported to cause abnormal insulin secretion in pancreatic β-cells [43]. Serum levels of lutein and β-carotene were found to be significantly negatively associated with diabetes-related markers (blood insulin, HOMA-IR) in both sexes, and they were also found to be negatively associated with FBG in females. Lutein and β-carotene might have a protective effect on pancreatic β-cells via an antioxidative effect.

There might be a possibility that serum total carotenoids are an alternative indicator of vegetable intake and that the associations between serum total carotenoid level and CVD markers observed in this study might be derived from some of the ingredients common in vegetables other than carotenoids. For example, dietary fibers in vegetables inhibit the increase of postprandial blood glucose levels and, thus, inhibit the progression of insulin resistance [44]. It is also thought that relatively low energy intake might suppress weight gain and the deterioration of CVD markers. Potassium, which is abundant in vegetables in general, and γ-aminobutyric acid, which is particularly abundant in tomatoes, can also decrease hypertension [45,46]. This suggests that these components are indirectly related to the significant relationship observed between carotenoids and blood pressure.

In addition to dietary habits, lifestyle habits, such as smoking, alcohol intake, and exercise, have a major impact on the development and progression of CVDs [47]. Therefore, these three items were added to the adjustment factors in the analysis. However, no significant change was observed in the results, suggesting that the relationships observed in this study are independent of the lifestyles mentioned above. Obesity also underlies many CVDs [48]. Therefore, BMI was added to the adjustment factors, but no significant change in the relationship between total carotenoids and CVD markers was observed in males or females, suggesting that these relationships are also independent of BMI. Thus, the associations that we found were probably due to the role of carotenoids and other ingredients of vegetables mentioned above.

However, these relationships were altered by age stratification (Table 2; Table 3). In young females, most of the relationships between carotenoids and CVD markers disappeared after adjusting for BMI. Thus, we speculate that obesity may have a negative impact on CVD markers, especially in young females, who are less likely to exhibit worsened premenopausal lipid-related markers.

In analyses with additional adjusted factors (lifestyle, BMI), a prominent number of significant relationships were observed between the serum total carotenoid levels and CVD markers in young males and middle-aged females. The levels of six CVD markers (BMI, baPWV, SBP, DBP, FBG, and TGs) were found to be higher in middle-aged males than in young. This suggests that the risk of CVDs starts increasing in young males. On the other hand, in females, menopause is known to alter lipid metabolism [49]. Hence, middle age, at which menopause occurs, might be the time when the deterioration of CVD markers begins through changes in the lipid metabolism. High dietary intake of vegetables (i.e., leading to a high serum concentration of total carotenoids) during this period may delay or prevent the exacerbation of CVD markers.

Our findings, obtained by stratifying sex and age, should act as useful references for a new intervention or prospective study to confirm the causal correlation that indicates when we should consume more carotenoid-rich vegetables.

### 4.3. Relationship between Individual Carotenoid Levels and Markers of CVDs

The physiological effects of carotenoids, other than their ability to scavenge singlet oxygen, have been reported to vary according to the type of carotenoid. Therefore, the association between individual carotenoids and CVD markers was investigated ( Table 4; Table 5).

Both β-carotene and lutein are found in many green and yellow vegetables; however, the relevant CVD markers with β-carotene and lutein differed. Such differences may also be influenced by the differences in the types of vegetables containing these carotenoids and in the content of vegetables [50]. However, it is presumed that these differences are also attributable to the distributions and physiological functions of these carotenoids in the body. For example, carotenes composed exclusively of carbon and hydrogen, such as β-carotene and lycopene, are found mainly in low-density lipoprotein (LDL) cholesterol particles in the blood, whereas xanthophylls containing hydroxyl groups in their structure, such as lutein, zeaxanthin, and β-cryptoxanthin, are distributed in relatively high proportions in HDL cholesterol particles [51]. Thus, the role of HDL cholesterol particles as a carrier of xanthophylls might be one of the reasons for the significant positive relationship between them rather than any physiological function of xanthophylls.

Lycopene, which is highly hydrophobic and weakly abundant in HDL cholesterol, is positively associated with HDL cholesterol in both males and females. It has been suggested that lycopene increases HDL cholesterol subtypes HDL-2 and HDL-3 and increases the HDL cholesterol in general by decreasing the activity of cholesteryl ester transfer protein in the blood and increasing the activity of lecithin–cholesterol acyltransferase (LCAT) in obese middle-aged individuals [52]. A mechanism of action has also been proposed whereby lycopene decreases LDL cholesterol by inhibiting 3-hydroxy-3-methylglutaryl-coenzyme A reductase [53]. Friedewald’s equation indicates that (LDL cholesterol) = (total cholesterol)–(HDL cholesterol)–(TGs)/5 when the blood TG level is 400 mg/dL or lower, suggesting an association between the reduction of LDL cholesterol and the elevation of HDL cholesterol. Another study showed that diets rich in lycopene actually increase the HDL cholesterol [54]. The positive relationship between lycopene and HDL cholesterol observed in this study may also be due to these physiological effects.

Serum levels of β-cryptoxanthin are significantly associated with multiple CVD markers in both males and females. This carotenoid, abundantly found in satsuma mandarin, was found to be associated with HOMA-IR in both sexes in the Mikkabi study [55] and was also associated with pulse velocity in both males and females [56]. In this study, it was hypothesized that β-cryptoxanthin has a protective effect against oxidative stress in the pathogenesis of insulin resistance. Additionally, β-cryptoxanthin has been reported to activate the nuclear factor erythroid 2-related factor 2 (Nrf2) pathway [57]. Sulforaphane is a typical activator of Nrf2 that has been found to improve insulin resistance in a mouse model [58]. In the present study, the serum concentrations of β-cryptoxanthin were significantly associated with diabetes-related markers in both sexes, but with baPWV only in females, although the subjects exhibited lower serum concentrations of β-cryptoxanthin compared to the population studied in the Mikkabi study.

The significant relationships between the serum concentration of β-cryptoxanthin and blood pressure or diabetes-related markers in females disappeared when we adopted BMI as the adjustment factor. This might suggest that obesity is a confounder for these associations. This is supported by previous reports, which showed that β-cryptoxanthin acts as an antagonist of peroxisome proliferation-activated receptor gamma (PPARγ) and may improve lipid metabolism and inhibit obesity via the inhibition of adipocyte hypertrophy [59,60].

Future examinations aimed at clarifying the causal relationships between each individual carotenoid and CVD markers on the basis of this study might reveal the carotenoid-rich vegetables that we should eat to prevent CVDs.

### 4.4. Study Limitations

As this was a cross-sectional study, it is unclear whether the intake of carotenoids directly improves CVD markers. Therefore, an analysis using an interventional study or a prospective cohort study is required to approach causality.

As this study was performed via a resident-based health examination, the participants were all Japanese living in a narrow area. Therefore, reproducibility should be confirmed in a different country and/or race.

Some associations between carotenoids and markers of glucose metabolism in males disappeared in all age categories, not just in a particular age group, suggesting that this disappearance may be mainly due to a reduction in statistical power due to a decrease in the number of subjects, especially among males. As mentioned in the Introduction, we conducted a stratified analysis by age despite the limitation of decreasing power of detection to obtain information on when we should eat carotenoid-rich vegetables to prevent CVDs. Increasing the number of subjects is expected to increase detection power in the future.

The BDHQ used in this study, whose validity has been verified several times [19], is often adopted in dietary surveys in Japan [61]. However, since this is a frequency survey, there may be a slight discrepancy between the actual vegetable intake and the return error caused by recall bias.

In this study, only those who voluntarily underwent medical examinations were enrolled because we used the medical examination data of the residents, suggesting a selection bias.

We analyzed many CVD markers in combination with carotenoids. It cannot be denied that the significant relationships observed in this study were errors due to multiple comparisons. However, to avoid picking up such errors, we focused on what changed in the series, rather than a single result.

## 5. Conclusions

Despite the limitations of this study, we found multiple associations between serum levels of carotenoids and CVD markers. Higher concentration of serum carotenoids in relatively healthy individuals is associated with better CVD markers. These associations continued to be significant even after adjusting for smoking habits, exercise habits, and alcohol intake, suggesting that these associations are independent of other lifestyle factors. In addition, there were more significantly suppressive associations between carotenoids and CVD markers in young males and middle-aged females, respectively, than in other age groups, in which the markers of CVDs begin to deteriorate for both sexes. Although these findings need to be confirmed using prospective cohort studies and interventional studies, they can be used as lifestyle guidance for disease prevention.

## Figures and Tables

**Figure 1 nutrients-12-02310-f001:**
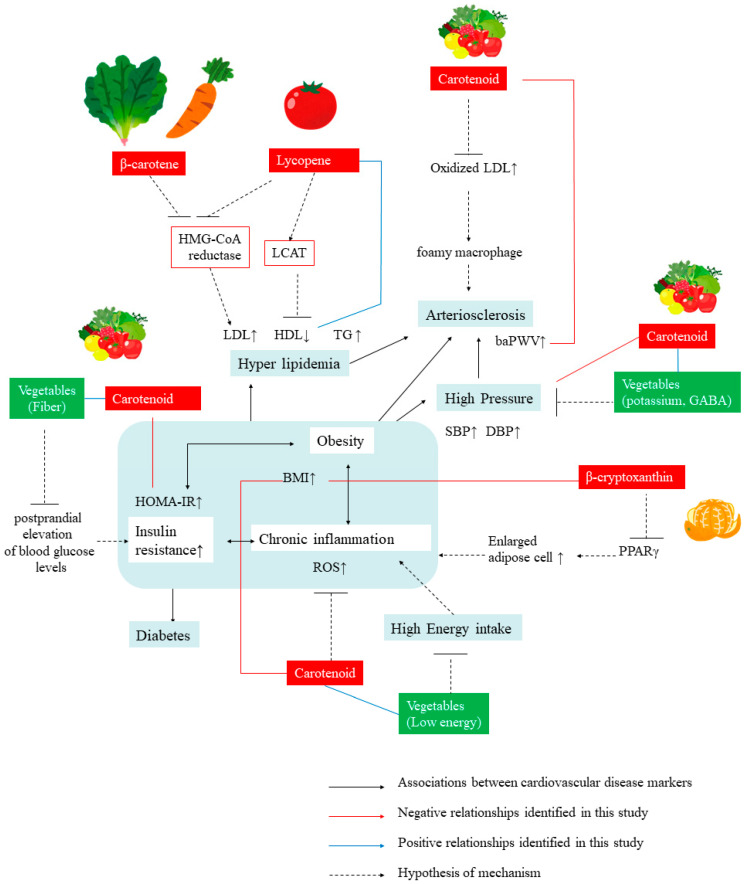
Diagram depicting roles played by carotenoids from vegetable intake on the prevention of cardiovascular diseases. BMI, body mass index; baPWV, brachial ankle pulse wave velocity; SBP, systolic blood pressure; DBP, diastolic blood pressure; HOMA-IR, Homeostatic Model Assessment for Insulin Resistance; FBG, fasting blood glucose; TG, triglyceride; HDL, high-density lipoprotein cholesterol; LDL, Low-density lipoprotein cholesterol; HOMA-IR, Homeostatic Model Assessment for Insulin Resistance; GABA, γ-aminobutyric acid; HMG-CoA reductase, 3-hydroxy-3-methylglutaryl-coenzyme A reductase; LCAT, Lecithin–cholesterol acyltransferase; PPARγ, Peroxisome proliferation-activated receptor gamma; ROS, reactive oxygen species.

**Table 1 nutrients-12-02310-t001:** Physiological characteristics of study subjects.

Measurement Item		Male				Female			
		All	Young (20–39 Years)	Middle-aged (40–59 Years)	Old (≥ 60 Years)	All	Young (20–39 Years)	Middle-aged (40–59 Years)	Old (≥ 60 years)
Number of samples	N	538	195	192	151	812	254	292	263
Basic markers	Age, year	48.2 ± 15.5	32 ± 5.1	48.8 ± 5.7 ^a^	68.2 ± 6.2 ^a, b^	49.8 ± 15.9	31.3 ± 5.3	49.6 ± 5.9 ^a^	68.1 ± 6.2 ^a, b^
	Current smoking, %	34	42.6	38	17.9 ^a, b^	11.8 ^***^	12.6 ^***^	17.8 ^***^	4.6 ^a, b ***^
	Habitual exercise, %	11	12.8	9.9	9.9	7.02 ^*^	6.3 ^*^	6.5	8.4
	Alcohol intake, g/day	23.5 ± 25.9	21 ± 25.06	26.07 ± 27.36	23.41 ± 24.72	4.91 ± 11.4 ^***^	5.17 ± 12.49 ^***^	6.76 ± 12.63 ^***^	2.51 ± 7.85 ^a, b ***^
	Antihypertensive use, %	17.1	0.5	13.0 ^a^	43.7 ^a, b^	18.0	0	9.6 ^a^	44.9 ^a, b^
Biomarkers	BMI, kg/m^2^	23.6 ± 3.37	23.39 ± 4	24.01 ± 2.9 ^a^	23.45 ± 3	22.00 ± 3.51 ^***^	20.83 ± 3.46 ^***^	22.6 ± 3.38 ^a ***^	22.91 ± 3.39 ^a, b *^
	baPWV, cm/s	1460.00 ± 353.00	1239.86 ± 167.98	1386.56 ± 195.81 ^a^	1837.81 ± 385.63 ^a, b^	1350.00 ± 346.00 ^***^	1068.73 ± 122.56 ^***^	1294.01 ± 248.90 ^a ***^	1685.72 ± 306.02 ^a, b ***^
	SBP, mmHg	126.00 ± 16.80	119.96 ± 13.37	124.68 ± 17.2 ^a^	133.74 ± 16.96 ^a, b^	118.00 ± 18.50 ^***^	107.86 ± 12.66 ^***^	117.31 ± 18.49 ^a ***^	128.57 ± 17.57 ^a, b **^
	DBP, mmHg	78.3 ± 12.5	74.41 ± 11.18	80.4 ± 13.22 ^a^	80.6 ± 11.83 ^a^	71.6 ± 11.3 ^***^	66.61 ± 9.33 ^***^	73.34 ± 12.35 a ^***^	74.55 ± 10.27 a ^***^
	HOMA-IR	1.05 ± 1.04	1.07 ± 1.03	1.13 ± 1.31	0.93 ± 0.55	0.98 ± 0.63	0.93 ± 0.53	0.93 ± 0.63 ^*^	1.08 ± 0.7 ^a, b *^
	Blood insulin, µU/mL	4.79 ± 3.32	5.12 ± 4.21	4.97 ± 3	4.13 ± 2.14 ^b^	4.73 ± 2.53	4.79 ± 2.57	4.46 ± 2.41	4.93 ± 2.53 ^b ***^
	FBG, mg/dL	85.8 ± 17.4	82.53 ± 15.4	86.02 ± 20.02 ^a^	89.86 ± 15.21 ^a, b^	82.1 ± 11.2 ^***^	77.48 ± 8.31 ^***^	82.02 ± 11.77 ^a **^	86.57 ± 11.25 ^a, b *^
	Triglyceride, mg/dL	121 ± 91.7	110.28 ± 93.59	135.17 ± 82.58 ^a^	115.36 ± 98.35 ^b^	76.5 ± 42.8 ^***^	62.93 ± 38.96 ^***^	77.46 ± 40.5 ^a ***^	88.47 ± 45.18 ^a, b **^
	HDL-cholesterol, mg/dL	59.9 ± 16.5	58.3 ± 15.42	60.18 ± 17.15	61.62 ± 16.87	70.3 ± 16.3 ^***^	69.7 ± 15.14 ^***^	72.61 ± 17.8 ^***^	68.46 ± 15.37 ^b ***^
BDHQ	Total vegetable, g/day	170.00 ± 108.00	150.74 ± 97.36	169.03 ± 99.15	196.3 ± 126.91 ^a^	180.00 ± 111.00 ^*^	153.64 ± 94.40	178.26 ± 100.42 ^a^	209.83 ± 129.2 ^a, b^
Carotenoids	Total carotenoid, µg/mL	1.1 ± 0.529	1.016 ± 0.448	1.131 ± 0.558	1.182 ± 0.572 ^a^	1.57 ± 0.713 ^***^	1.363 ± 0.607 ^***^	1.573 ± 0.684 ^a ***^	1.777 ± 0.781 ^a, b ***^
	Lutein, µg/mL	0.287 ± 0.136	0.236 ± 0.096	0.294 ± 0.129 ^a^	0.345 ± 0.16 ^a, b^	0.333 ± 0.15 ^***^	0.265 ± 0.108 ^**^	0.332 ± 0.144 ^a **^	0.403 ± 0.16 ^a, b ***^
	Zeaxanthin, µg/mL	0.061 ± 0.022	0.059 ± 0.021	0.064 ± 0.022	0.059 ± 0.023	0.0618 ± 0.0239	0.06 ± 0.023	0.063 ± 0.022	0.063 ± 0.026
	β-Cryptoxanthin, µg/mL	0.104 ± 0.059	0.093 ± 0.044	0.099 ± 0.056	0.123 ± 0.073 ^a, b^	0.163 ± 0.102 ^***^	0.129 ± 0.062 ^***^	0.158 ± 0.099 ^a ***^	0.203 ± 0.121 ^a, b ***^
	α-Carotene, µg/mL	0.127 ± 0.147	0.120 ± 0.125	0.138 ± 0.178	0.122 ± 0.127	0.183 ± 0.149 ^***^	0.17 ± 0.153 ^***^	0.183 ± 0.136 ^a ***^	0.194 ± 0.157 ^a ***^
	β-Carotene, µg/mL	0.279 ± 0.258	0.232 ± 0.207	0.278 ± 0.279	0.343 ± 0.277 ^a, b^	0.561 ± 0.404 ^***^	0.434 ± 0.317 ^***^	0.544 ± 0.362 ^a ***^	0.705 ± 0.475 ^a, b ***^
	Lycopene, µg/mL	0.247 ± 0.142	0.276 ± 0.142	0.258 ± 0.137	0.194 ± 0.134 ^a, b^	0.27 ± 0.147 ^***^	0.304 ± 0.137 **	0.292 ± 0.15 ^**^	0.21 ± 0.135 ^a, b^

Median values were significantly different from males (Mann–Whitney *U* test): * *p* < 0.05, ** *p* < 0.01, *** *p* < 0.001. Median values were significantly different from the young group (Kruskal–Willis test; post hoc: Bonferroni): ^a^
*p* < 0.05. Median values were significantly different from the middle-aged group (Kruskal–Willis test; post hoc: Bonferroni): ^b^
*p* < 0.05. BMI, body mass index; baPWV, brachial ankle pulse wave velocity; SBP, systolic blood pressure; DBP, diastolic blood pressure; HOMA-IR, Homeostatic Model Assessment for Insulin Resistance; FBG, fasting blood glucose; TG, triglyceride; HDL, high-density lipoprotein; BDHQ, brief-type self-administered diet history questionnaire.

**Table 2 nutrients-12-02310-t002:** Multiple linear regression analysis of the three age groups showing association between cardiovascular disease markers and total blood carotenoid concentrations in males.

Biomarkers	Pattern 1				Pattern 2				Pattern 3			
	All	20–39	40–59	60–	All	20–39	40–59	60–	All	20–39	40–59	60–
BMI	−0.060	−0.037	−0.179 ^*^	0.008	−0.054	−0.003	−0.218 ^*^	0.032				
baPWV	−0.114 ^***^	−0.259 ^**^	−0.169 ^*^	−0.100	−0.121 ^***^	−0.248 ^**^	−0.168 ^*^	−0.092	−0.119 ^***^	−0.248 ^**^	−0.162 ^*^	−0.091
SBP	−0.177 ^***^	−0.140	−0.197 ^*^	−0.167	−0.178 ^***^	−0.158	−0.206 ^*^	−0.155	−0.163 ^***^	−0.157	−0.148	−0.157
DBP	−0.157 ^**^	−0.073	−0.147	−0.237 ^**^	−0.135 ^**^	−0.056	−0.130	−0.232 ^**^	−0.119 ^*^	−0.055	−0.077	−0.235 ^**^
HOMA−IR	−0.129 ^**^	−0.164	−0.047	−0.095	−0.135 ^**^	−0.139	−0.052	−0.098	−0.102 ^*^	−0.186 ^**^	0.039	−0.117
Insulin	−0.104 ^*^	−0.158	−0.072	−0.096	−0.109 ^*^	−0.156	−0.089	−0.096	−0.080 ^*^	−0.139 ^*^	0.014	−0.123
FBG	−0.055	−0.202 ^**^	0.004	−0.025	−0.025	−0.182 ^*^	0.006	−0.006	−0.007	−0.182 ^*^	0.033	0.017
TG	−0.212 ^***^	−0.252 ^**^	−0.182 ^*^	−0.262 ^**^	−0.201 ^***^	−0.230 ^*^	−0.201 ^*^	−0.257 ^**^	−0.187 ^***^	−0.229 ^**^	−0.158	−0.262 ^**^
HDL cholesterol	0.172 ^***^	0.191 ^*^	0.212 ^*^	0.169	0.220 ^***^	0.180 ^*^	0.297 ^***^	0.200 ^*^	0.198 ^***^	0.179 ^*^	0.264 ^**^	0.191 *

Yellow background color: relationship become non-significant by stratifying by age categories. Green background color: relationship become significant by stratifying by age categories. * *p* < 0.05, ** *p* < 0.01, *** *p* < 0.001. Significant values are shown in red. Pattern 1: calculated using multiple linear regression analysis after adjusting for age, antihypertensive drug use, and vitamins A, C, and E in the blood. Pattern 2: calculated using multiple linear regression analysis after adjusting for age, antihypertensive drug use, smoking habits, exercise habits, alcohol intake, and vitamins A, C, and E in the blood. Pattern 3: calculated using multiple linear regression analysis after adjusting for age, antihypertensive drug use, smoking habits, exercise habits, alcohol intake, BMI, and vitamins A, C, and E in the blood. BMI, body mass index; baPWV, brachial ankle pulse wave velocity; SBP, systolic blood pressure; DBP, diastolic blood pressure; HOMA-IR, Homeostatic Model Assessment for Insulin Resistance; FBG, fasting blood glucose; TG, triglyceride; HDL, high-density lipoprotein.

**Table 3 nutrients-12-02310-t003:** Multiple linear regression analysis of the three age groups showing association between cardiovascular disease markers and total blood carotenoid concentrations in females.

Biomarkers	Pattern 1				Pattern 2				Pattern 3			
	All	20–39	40–59	60–	All	20–39	40–59	60–	All	20–39	40–59	60–
BMI	−0.235 ^***^	−0.233 ^***^	−0.268 ^***^	−0.243 ^***^	−0.238 ^***^	−0.254 ^***^	−0.266 ^***^	−0.239 ^***^				
baPWV	−0.065 ^**^	−0.137 ^*^	−0.090	−0.077	−0.088 ^***^	−0.135 ^*^	−0.128 ^*^	−0.101	−0.091 ^***^	−0.101	−0.112	−0.134 ^*^
SBP	−0.119 ^***^	−0.186 ^**^	−0.176 ^**^	−0.047	−0.134 ^***^	−0.190 ^**^	−0.195 ^**^	−0.075	−0.071 ^*^	−0.067	−0.126 ^*^	−0.048
DBP	−0.069	−0.124	−0.122 ^*^	−0.034	−0.066	−0.112	−0.159 ^*^	−0.032	−0.006	−0.038	−0.102	0.017
HOMA−IR	−0.228 ^***^	−0.231 ^***^	−0.254 ^***^	−0.184 ^**^	−0.252 ^***^	−0.260 ^***^	−0.280 ^***^	−0.203 ^**^	−0.163 ^***^	−0.155 ^*^	−0.171 ^**^	−0.102
Insulin	−0.228 ^***^	−0.221 ^***^	−0.275 ^***^	−0.172 ^*^	−0.259 ^***^	−0.249 ^***^	−0.292 ^***^	−0.199 ^**^	−0.158 ^***^	−0.141 ^*^	−0.151 ^**^	−0.096
FBG	−0.134 ^***^	−0.142 *	−0.200 ^***^	−0.075	−0.132 ^***^	−0.142 ^*^	−0.189 ^***^	−0.098	−0.102 ^**^	−0.102	−0.159 ^**^	−0.054
TG	−0.162 ^***^	−0.033	−0.234 ^***^	−0.242 ^***^	−0.150 ^***^	−0.028	−0.229 ^***^	−0.258 ^***^	−0.096 ^*^	0.016	−0.156 ^*^	−0.218 ^**^
HDL cholesterol	0.303 ^***^	0.243 ^***^	0.281 ^***^	0.329 ^***^	0.319 ^***^	0.252 ^***^	0.291 ^***^	0.348 ^***^	0.247 ^***^	0.162 ^*^	0.208 ^***^	0.283 ^***^

Yellow background color: relationship become non-significant by stratifying by age categories. Green background color: relationship become significant by stratifying by age categories. * *p* < 0.05, ** *p* < 0.01, *** *p* < 0.001. Significant values are shown in red. Pattern 1: calculated using multiple linear regression analysis after adjusting for age, antihypertensive drug use, and vitamins A, C, and E in the blood. Pattern 2: calculated using multiple linear regression analysis after adjusting for age, antihypertensive drug use, smoking habits, exercise habits, alcohol intake, and vitamins A, C, and E in the blood. Pattern 3: calculated using multiple linear regression analysis after adjusting for age, antihypertensive drug use, smoking habits, exercise habits, alcohol intake, BMI, and vitamins A, C, and E in the blood. BMI, body mass index; baPWV, brachial ankle pulse wave velocity; SBP, systolic blood pressure; DBP, diastolic blood pressure; HOMA-IR, Homeostatic Model Assessment for Insulin Resistance; FBG, fasting blood glucose; TG, triglyceride; HDL, high-density lipoprotein.

**Table 4 nutrients-12-02310-t004:** Multiple linear regression analysis showing association between cardiovascular disease markers and each carotenoid’s concentration in the blood in males.

Biomarkers	Pattern 1						Pattern 2						Pattern 3					
	Total Carotenoid (Repeated)	Lutein	Zeaxanthin	β-Cryptoxanthin	β-Carotene	Lycopene	Total Carotenoid (Repeated)	Lutein	Zeaxanthin	β-Cryptoxanthin	β-Carotene	Lycopene	Total Carotenoid (Repeated)	Lutein	Zeaxanthin	β−Cryptoxanthin	β−Carotene	Lycopene
BMI	−0.06	−0.087	0.012	−0.044	−0.05	0.026	−0.054	−0.083	0.014	−0.046	−0.037	0.042						
baPWV	−0.114 ^***^	−0.002	−0.013	−0.05	−0.112 ^***^	−0.079 ^**^	−0.121 ^***^	0.002	−0.012	−0.056	−0.124 ^***^	−0.081 ^**^	−0.119 ^***^	0.004	−0.011	−0.055	−0.123 ^***^	−0.081 ^**^
SBP	−0.177 ^***^	−0.072	−0.012	−0.137 ^***^	−0.204 ^***^	−0.117 ^**^	−0.178 ^***^	−0.075	−0.011	−0.159 ^***^	−0.214 ^***^	−0.119 ^**^	−0.163 ^***^	−0.055	−0.011	−0.148 ^***^	−0.206 ^***^	−0.128 ^**^
DBP	−0.157 ^**^	−0.003	0.045	−0.180 ^***^	−0.224 ^***^	−0.078	−0.135 ^**^	−0.010	0.042	−0.165 ^***^	−0.185 ^***^	−0.063	−0.119 *	0.014	0.044	−0.153 ^***^	−0.176 ^***^	−0.072
HOMA−IR	−0.129 ^**^	−0.166 ^***^	−0.043	−0.094 ^*^	−0.137 ^**^	0.001	−0.135 ^**^	−0.158 ^***^	−0.035	−0.121 ^*^	−0.173 ^***^	0.001	−0.102 ^*^	−0.148 ^***^	−0.054	−0.122 ^**^	−0.142 ^***^	−0.074 ^*^
Insulin	−0.104 ^*^	−0.191 ^***^	−0.06	−0.096 ^*^	−0.126 ^**^	−0.024	−0.109 ^*^	−0.176 ^***^	−0.046	−0.129 ^**^	−0.171 ^***^	−0.028	−0.080 ^*^	−0.129 ^***^	−0.057	−0.105 ^**^	−0.150 ^***^	−0.054
FBG	−0.055	−0.047	0.028	−0.041	−0.062	−0.01	−0.025	−0.054	0.021	−0.027	−0.042	0.002	−0.007	−0.033	0.021	−0.014	−0.032	−0.006
TG	−0.212 ^***^	−0.021	0.079	−0.118 ^**^	−0.219 ^***^	0.01	−0.201 ^***^	−0.022	0.063	−0.075	−0.169 ^***^	0.019	−0.187 ^***^	0	0.062	−0.063	−0.160 ^***^	0.009
HDL cholesterol	0.172 ^***^	0.319 ^***^	0.233 ^***^	0.155 ^***^	0.041	0.161 ^***^	0.220 ^***^	0.273 ^***^	0.191 ^***^	0.202 ^***^	0.091 ^*^	0.183 ^***^	0.198 ^***^	0.248 ^***^	0.190 ^***^	0.186 ^***^	0.078	0.194 ^***^

Green background color: negatively associated. Yellow background color: positively associated * *p* < 0.05, ** *p* < 0.01, *** *p* < 0.001. (**a**) Pattern 1: calculated using multiple linear regression analysis after adjusting for age and antihypertensive drug use. Blood concentrations of vitamins A, C and E were also adopted as adjustment factors for the evaluation of total carotenoids. Pattern 2: calculated using multiple linear regression analysis after adjusting for age, antihypertensive drug use, smoking habits, exercise habits, and alcohol intake. Blood concentrations of vitamins A, C and E were also adopted as adjustment factors for the evaluation of total carotenoids. (**b**) Pattern 3: calculated using multiple linear regression analysis after adjusting for age, antihypertensive drug use, smoking habits, exercise habits, alcohol intake, and BMI. Blood concentrations of vitamins A, C and E were also adopted as adjustment factors for the evaluation of total carotenoids. BMI, body mass index; baPWV, brachial ankle pulse wave velocity; SBP, systolic blood pressure; DBP, diastolic blood pressure; HOMA−IR, Homeostatic Model Assessment for Insulin Resistance; FBG, fasting blood glucose; TG, triglyceride; HDL, high-density lipoprotein.

**Table 5 nutrients-12-02310-t005:** Multiple linear regression analysis showing association between cardiovascular disease markers and each carotenoid’s concentration in the blood in females.

Biomarker	Pattern 1						Pattern 2						Pattern 3					
	Total Carotenoid (Repeated)	Lutein	Zeaxanthin	β-Cryptoxanthin	β-Carotene	Lycopene	Total Carotenoid (repeated)	Lutein	Zeaxanthin	β-Cryptoxanthin	β-Carotene	Lycopene	Total Carotenoid (Repeated)	Lutein	Zeaxanthin	β-Cryptoxanthin	β-Carotene	Lycopene
BMI	−0.235 ^***^	−0.250 ^***^	−0.125 ^***^	−0.137 ^***^	−0.229 ^***^	−0.049	−0.238 ^***^	−0.251 ^***^	−0.121 ^***^	−0.144 ^***^	−0.238 ^***^	−0.053						
baPWV	−0.065 ^**^	−0.018	0.001	−0.047 ^*^	−0.090 ^***^	−0.054 ^*^	−0.088 ^***^	−0.039	−0.015	−0.055 ^*^	−0.110 ^***^	−0.068 ^**^	−0.091 ^***^	−0.040	−0.015	−0.057 ^*^	−0.115 ^***^	−0.067 ^**^
SBP	−0.119 ^***^	−0.090 ^**^	−0.019	−0.080 ^*^	−0.152 ^***^	−0.064 ^*^	−0.134 ^***^	−0.102 ^**^	−0.024	−0.082 ^*^	−0.168 ^***^	−0.056	−0.071 ^*^	−0.034	0.01	−0.040	−0.106 ^**^	−0.039
DBP	−0.069	−0.032	0.064	−0.038	−0.131 ^***^	0.031	−0.066	−0.044	0.045	−0.029	−0.119 ^**^	0.034	−0.006	0.022	0.076 ^*^	0.01	−0.063	0.05
HOMA−IR	−0.228 ^***^	−0.243 ^***^	−0.134 ^***^	−0.077 ^*^	−0.249 ^***^	−0.032	−0.252 ^***^	−0.246 ^***^	−0.132 ^***^	−0.087 ^*^	−0.278 ^***^	−0.043	−0.163 ^***^	−0.141 ^***^	−0.080 ^*^	−0.024	−0.183 ^***^	−0.021
Insulin	−0.228 ^***^	−0.234 ^***^	−0.128 ^***^	−0.080 ^*^	−0.247 ^***^	−0.029	−0.259 ^***^	−0.236 ^***^	−0.123 ^***^	−0.092 ^*^	−0.282 ^***^	−0.041	−0.158 ^***^	−0.118 ^***^	−0.065 ^*^	−0.024	−0.176 ^***^	−0.017
FBG	−0.134 ^***^	−0.192 ^***^	−0.105 ^**^	−0.063	−0.180 ^***^	−0.017	−0.132 ^***^	−0.214 ^***^	−0.111 ^***^	−0.047	−0.164 ^***^	−0.025	−0.102 ^**^	−0.180 ^***^	−0.095 ^**^	−0.018	−0.128 ^***^	−0.012
TG	−0.162 ^***^	−0.123 ^***^	−0.057	−0.092 ^*^	−0.196 ^***^	0.025	−0.150 ^***^	−0.103 ^**^	−0.049	−0.049	−0.182 ^***^	0.023	−0.096 ^*^	−0.036	−0.019	−0.010	−0.128 ^***^	0.029
HDL cholesterol	0.303 ^***^	0.462 ^***^	0.376 ^***^	0.159 ^***^	0.207 ^***^	0.189 ^***^	0.319 ^***^	0.435 ^***^	0.345 ^***^	0.170 ^***^	0.240 ^***^	0.191 ^***^	0.247 ^***^	0.365 ^***^	0.307 ^***^	0.119 ^***^	0.162 ^***^	0.170 ^***^

Green background color: negatively associated. Yellow background color: positively associated. * *p* < 0.05, ** *p* < 0.01, *** *p* < 0.001. (**a**) Pattern 1: calculated using multiple linear regression analysis after adjusting for age and antihypertensive drug use. Blood concentrations of vitamins A, C and E were also adopted as adjustment factors for the evaluation of total carotenoids; Pattern 2: calculated using multiple linear regression analysis after adjusting for age, antihypertensive drug use, smoking habits, exercise habits, and alcohol intake. Blood concentrations of vitamins A, C and E were also adopted as adjustment factors for the evaluation of total carotenoids. (**b**) Pattern 3: calculated using multiple linear regression analysis after adjusting for age, antihypertensive drug use, smoking habits, exercise habits, alcohol intake, and BMI. Blood concentrations of vitamins A, C and E were also adopted as adjustment factors for the evaluation of total carotenoids. BMI, body mass index; baPWV, brachial ankle pulse wave velocity; SBP, systolic blood pressure; DBP, diastolic blood pressure; HOMA-IR, Homeostatic Model Assessment for Insulin Resistance; FBG, fasting blood glucose; TG, triglyceride; HDL, high-density lipoprotein.

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
