# Peer review of "Association between Biomarkers of Cardiovascular Diseases and the Blood Concentration of Carotenoids among the General Population without Apparent Illness"

_nutrients, 2020, doi:10.3390/nu12082310_

Round 1

Reviewer 1 Report

This paper determines the relationship between selected CVD risk factors and blood concentrations of carotenoids.

This study is not particularly novel. To date, many papers have been published that deal with such topics. In addition, as the authors have mentioned, the cross-sectional study has its limitations and does not allow to clearly answer the research questions posed. On the other hand, even intervention studies using animal models or human studies have some limitations.

This work falls within the scope of the journal Nutrients.
Moreover, I think that to broaden / confirm the current state of knowledge, work such as this is a contribution in the field of food and its impact on health.

This work is consistent and transparent. Research methods, results and discussion have been described properly.

This work requires minor changes:

Line 23-25. Abbreviations are needed, eg. "pulse wave velocity", "systolic blood pressure", "Homeostatic Model Assessment for Insulin Resistance", "high-density lipoprotein cholesterol" (HOMA-IR, HDL etc.).

Table 2 and 3 (p. 8). It requires correction (horizontal page layout).

Introduction. It is worth mentioning about other recent works, e.g. reviews, which undertook similar topics (relationship between carotenoids and CVD).

eg. https://www.sciencedirect.com/science/article/abs/pii/S1756464617305133 and https://www.ncbi.nlm.nih.gov/pmc/articles/PMC4321000/

I recommend this work for publication.

Author Response

Response to Reviewer 1 Comments

Comments from Reviewer 1;

This paper determines the relationship between selected CVD risk factors and blood concentrations of carotenoids.

This study is not particularly novel. To date, many papers have been published that deal with such topics. In addition, as the authors have mentioned, the cross-sectional study has its limitations and does not allow to clearly answer the research questions posed. On the other hand, even intervention studies using animal models or human studies have some limitations.

This work falls within the scope of the journal Nutrients.

Moreover, I think that to broaden / confirm the current state of knowledge, work such as this is a contribution in the field of food and its impact on health.

This work is consistent and transparent. Research methods, results and discussion have been described properly.

Response: Thank you very much for your comments on our study.

This work requires minor changes:

Point 1: Line 23-25. Abbreviations are needed, eg. "pulse wave velocity", "systolic blood pressure", "Homeostatic Model Assessment for Insulin Resistance", "high-density lipoprotein cholesterol" (HOMA-IR, HDL etc.).

Response 1: Thank you for your comment. The following revision has been made.

(Page 1, Lines 20-25 in the revised version)

“We then evaluated the relationship between the serum concentrations of six major carotenoids as well as their total and nine CVD markers (i.e., body mass index [BMI], pulse wave velocity [PWV], systolic blood pressure [SBP], diastolic blood pressure [DBP], Homeostatic Model Assessment of Insulin Resistance [HOMA-IR], blood insulin, fasting blood glucose [FBG], triglycerides [TGs], and high-density lipoprotein [HDL] cholesterol) using multiple regression analysis.”

Point 2: Table 2 and 3 (p. 8). It requires correction (horizontal page layout).

Response 2: Thank you for your comment. We made Table 2 and 3 as horizontal page layout (landscape mode). Is this table intended to be horizontal on A4 vertical paper? If so, should Tables 4 and 5 be modified in the same way? It's still unadjusted because I wasn't sure of the intent. If my perceptions seem correct, I will try to correct them.

Point 3: Introduction. It is worth mentioning about other recent works, e.g. reviews, which undertook similar topics (relationship between carotenoids and CVD).

  1. https://www.sciencedirect.com/science/article/abs/pii/S1756464617305133 and https://www.ncbi.nlm.nih.gov/pmc/articles/PMC4321000/

Response 3: Thank you for your comment. This point has been raised by another reviewer as well. The following revisions have been made.

(Page 2, Lines 52-55 in the revised version)

“Various epidemiological studies suggest that higher intake of vegetables and fruits decreases the risk of CVDs by lowering the blood pressure, reducing the levels of proinflammatory cytokines and inflammatory markers, and improving insulin sensitivity [15], as noted earlier.”

I recommend this work for publication.

Response: Again, thank you very much for your comment on our paper. We believe that we have appropriately responded to all of the above comments. We also made many changes according to the other reviewers’ suggestions. All of changes which we made were able to be checked by the tracked changes of the revised manuscript.

Reviewer 2 Report

The Authors have investigated novel insights into the relationships between serum carotenoid levels and cardiovascular disease markers in healthy individuals by identifying differences in relation to sex and age.

Although this study is characterized by a large number of evaluations on elevated number of individuals without apparent illness, its weakness is represented by the absence of any mechanistic demonstrations. Thus, the Authors should at least propose the possible mechanistic hypotheses adding evaluations of circulating biomarkers of early endothelial dysfunction, which is the starting point of the atherosclerotic process and accompanies the progression of CVD.

So far, numerous studies have been conducted considering traditional, also defined “classic”, endothelial circulating markers, including soluble adhesion molecules, such as E-selectin, ICAM-1 and VCAM-1, as well as molecules involved in the coagulation pathway, in particular von Willebrand factor (vWF) and soluble thrombomodulin, but also inflammatory markers such as interleukins IL-6, IL-8, IL-12 and high-sensitivity C-reactive protein (hsCRP) (Leite AR et al,  Novel Biomarkers for Evaluation of Endothelial Dysfunction. Angiology 2020;71:397–410; Goncharov N V et al, Markers and Biomarkers of Endothelium: When Something Is Rotten in the State. Oxid Med Cell Longev 2017; Defagó MD et al, J Clin Hypertens 2014;16:907–13).

In the Discussion session they cite other studies in which the mechanisms have already been hypothesized, but this study should be extended to evaluate the association between vegetable and fruit consumption, carotenoids plasma levels, modulation of biomarkers of early endothelial dysfunction and then markers of CVD. This approach might support a real current need of new more validated, appropriate, and reliable diagnostic and therapeutic biomarkers useful both to diagnose endothelial dysfunction at an earlier stage and its relation with diet.

Minor:

Title: Association between Biomarkers of Cardiovascular  Diseases and the Blood Concentrations of (Antioxidants, should be corrected as follow) Carotenoids Among the General Population without Apparent Illness.

Abstract: “…start eating carotenoids” should be changed with “consumption of various vegetables and fruits and then carotenoids ingestion”, or something like that.

Discussion: there are several sentences that might be revised, for example “Even in healthy individuals, we found a statistically significant relationship between serum carotenoids and various markers before various markers appeared abnormal, and these relationships varied according to the sex and age categories (Figure 1)”, and “Higher intake of carotenoid-rich vegetables in relatively healthy individuals is associated with better CVD markers”.

Author Response

Response to Reviewer 2 Comments

Comments from Reviewer 2;

The Authors have investigated novel insights into the relationships between serum carotenoid levels and cardiovascular disease markers in healthy individuals by identifying differences in relation to sex and age.

Although this study is characterized by a large number of evaluations on elevated number of individuals without apparent illness, its weakness is represented by the absence of any mechanistic demonstrations. Thus, the Authors should at least propose the possible mechanistic hypotheses adding evaluations of circulating biomarkers of early endothelial dysfunction, which is the starting point of the atherosclerotic process and accompanies the progression of CVD.

So far, numerous studies have been conducted considering traditional, also defined “classic”, endothelial circulating markers, including soluble adhesion molecules, such as E-selectin, ICAM-1 and VCAM-1, as well as molecules involved in the coagulation pathway, in particular von Willebrand factor (vWF) and soluble thrombomodulin, but also inflammatory markers such as interleukins IL-6, IL-8, IL-12 and high-sensitivity C-reactive protein (hsCRP) (Leite AR et al, Novel Biomarkers for Evaluation of Endothelial Dysfunction. Angiology 2020;71:397–410; Goncharov N V et al, Markers and Biomarkers of Endothelium: When Something Is Rotten in the State. Oxid Med Cell Longev 2017; Defagó MD et al, J Clin Hypertens 2014;16:907–13).

In the Discussion session they cite other studies in which the mechanisms have already been hypothesized, but this study should be extended to evaluate the association between vegetable and fruit consumption, carotenoids plasma levels, modulation of biomarkers of early endothelial dysfunction and then markers of CVD. This approach might support a real current need of new more validated, appropriate, and reliable diagnostic and therapeutic biomarkers useful both to diagnose endothelial dysfunction at an earlier stage and its relation with diet.

Response: Thank you for your comment. As you have pointed out, the circulating biomarkers of early endothelial dysfunction are very important markers that need to be evaluated in the future. The following discussion has been added.

(Page 16-17, Lines 314-322 in the revised version)

“In the present study, we did not evaluate any circulating biomarkers of early endothelial dysfunction, which is the starting point of the atherosclerotic process and accompanies the progression of CVDs [36]. We suggest that tomatoes can help prevent atherosclerosis by inhibiting vascular endothelial dysfunction in a primitive animal experiment [37]. Tsitsimpikou et al. suggested that continuous intake of tomato juice significantly improves the inflammation status and endothelial dysfunction [38]. In the future, we aim to elucidate the mechanism in detail by examining the relationship between internal carotenoid levels and markers such as E-selectin, ICAM-1, VCAM-1, and von Willebrand Factor as well as inflammatory markers.”

Minor:

Point 1: Title: Association between Biomarkers of Cardiovascular Diseases and the Blood Concentrations of (Antioxidants, should be corrected as follow) Carotenoids Among the General Population without Apparent Illness.

Response 1: Thank you for your comment. The title has been revised as follows.

(Title in the revised version)

“Association between Biomarkers of Cardiovascular Diseases and the Blood Concentration of Carotenoids among the General Population without Apparent Illness”

Point 2: Abstract: “…start eating carotenoids” should be changed with “consumption of various vegetables and fruits and then carotenoids ingestion”, or something like that.

Response 2: Thank you for your comment. However, because we wanted to focus on vegetables in this study, we excluded fruits. The following revision has been made.

(Page 1, Lines 16-18 in the revised version)

“Several studies have demonstrated that carotenoid-rich vegetables are useful against cardiovascular diseases (CVDs). However, it is still unclear when a healthy population should start eating those vegetables to prevent CVDs.”

Point 3: Discussion: there are several sentences that might be revised, for example “Even in healthy individuals, we found a statistically significant relationship between serum carotenoids and various markers before various markers appeared abnormal, and these relationships varied according to the sex and age categories (Figure 1)”, and “Higher intake of carotenoid-rich vegetables in relatively healthy individuals is associated with better CVD markers”.

Response 3: Thank you for your comment. A similar point has been raised by another reviewer as well. Some sentences in the discussion were revised as follows.

(Page 14, Lines 260-262 in the revised version)

“Even in healthy individuals without apparent illness, we found a statistically significant relationship between serum carotenoids and various CVD markers, and some of these relationships varied according to the sex and age categories.”

(Page 19, Lines 448-449 in the revised version)

“Higher concentration of serum carotenoids in relatively healthy individuals is associated with better CVD markers.”

If there are any further corrections that need to be made, please let me know.

Response: Again, thank you very much for your comment on our paper. We believe that we have appropriately responded to all of the above comments. We also made many changes according to the other reviewers’ suggestions. All of changes which we made were able to be checked by the tracked changes of the revised manuscript.

Reviewer 3 Report

This study evaluates cross-sectional relationships between serum carotenoids and CVD markers in Japanese adults. I believe that there are some important points that need to be addressed before this paper is considered for publication.

  1. The Results section should be more concise and have a better flow so it can provide a cohesive story to the reader. I provide some specific recommendations below:
    1. In Section 3.1, please make sure that you avoid lengthy detailed descriptions of your statistics, which can be tiring to the reader, and focus on a few striking points. Also, make sure that your text corresponds to the results in Table 1. For example, in page 4, line 141 it is mentioned that BMI was higher in middle-aged males compared to other age groups but the table indicates that this is the case only compared to the younger group.
    2. When referring to regression coefficients please prefer words like association or relationship rather than correlation which mostly refers to correlation coefficients.
    3. In Section 3.4.1, please provide a sentence on what are the associations between total carotenoids and CVD markers before comparing them with other exposures like b-carotene and b-cryptoxanthin.
    4. As text under some subheadings is scarce (e.g. page 14, lines 12 and 15), I’d suggest not using subheadings for every model specification. It might make more sense to focus your results on fully adjusted models.
    5. Page 7, line 3: did you consider including more confounders in this regression?
  2. The writing style of the Discussion also needs to be significantly improved. Here are some specific points:
    1. It is important not to repeat your results in the discussion but use this space to compare with relevant up-to-date literature. For example, in page 17, lines 91-108 are just a repetition of results.
    2. In sections 4.3 and 4.4 you don’t need to discuss every model specification separately.
    3. Section 4.4.1 provides a large description of potential mechanisms of identified relationships. This would be clearer to the reader if it was provided in a separate section with a relevant subheading.
    4. In Section 4.2, the authors refer to limitations of food frequency questionnaires used in the study by Deborah et al. (please use author last name here) but it is not clear how the dietary assessment method used in their analysis was superior. Also, there are more recent studies investigating associations between plasma carotenoid and vegetable intake, including systematic reviews e.g. https://doi.org/10.1017/S0007114515003165
    5. Surprised not to see any comments on the external validity of the study and the potential problems with multiple testing in Section 4.5
  3. Methods
    1. It is not clear to me why authors proceeded with age stratification, especially given the potential lack of power in these groups (sample size 151-292). Was there strong indication of effect modification with age? I understand that the authors imply that age-stratification would indicate when carotenoids need to be consumed to prevent CVD but I doubt that a cross-sectional analysis can answer this question.
    2. Please give more details on the BDHQ used, including the time frame of consumption it corresponds, if it was quantitative, created or adapted for local diets, etc.
    3. Also, did the BDHQ provide information on dietary intake of other groups, such as fruit? Did you consider adjusting for other dietary confounders? Please specify the measurement units of vegetable and alcohol intake.
    4. Were height and weight self-reported or measured? If measured, how?
    5. Does smoking habits refer only to current smoking and to all tobacco products?
    6. page 3, line 127-128s: Please give justification for why you adjusted for serum/plasma vitamins.
    7. Page 3, line 116: are the results of this correlation analysis presented anywhere?
    8. Page 3, line 118: please mention the covariates you used in this regression.
  4. Introduction:
    1. I am not fully aware of all the relevant literature but it seems that authors tend to cite older papers. Isn’t there any more recent and robust work to be cited? See for example, this seminal meta-analysis by Aune https://doi.org/10.1093/ije/dyw319
    2. Please clearly provide the aim of this study in the last paragraph of the introduction.
  5. Conclusions:
    1. Page 19, lines 202-203: You imply that your study showed associations between vegetable intake and CVD markers which is not the case. Please make sure that the conclusion reflects your results.
    2. Page 19, lines 205-207: It is not clear which associations you are referring to, I assume this is about associations between total carotenoids and CVD markers. It is a bit misleading to focus on these age groups as associations don’t seem to be stronger in these groups compared to others, e.g. middle-aged males and young females.

Author Response

Response to Reviewer 3 Comments

Comments from Reviewer 3;

This study evaluates cross-sectional relationships between serum carotenoids and CVD markers in Japanese adults. I believe that there are some important points that need to be addressed before this paper is considered for publication.

  1. The Results section should be more concise and have a better flow so it can provide a cohesive story to the reader. I provide some specific recommendations below:

Response: Thank you for your insightful comments. These comments helped us significantly improve the paper.

    1. In Section 3.1, please make sure that you avoid lengthy detailed descriptions of your statistics, which can be tiring to the reader, and focus on a few striking points.

Response: Thank you for your comment. The following revisions have been made in Section 3.1. We also made our description simple throughout the manuscript.

(Page 4, Lines 139-147 in the revised version)

“Table 1 shows the mean values of all measurements stratified by sex and age categories, and many differences were found among sex and age groups. Most of the CVD markers worsened with age. However, two markers (BMI and TG) showed the highest value in middle-aged males and two markers (blood insulin and HDL) began to worsen significantly in older females. The vegetable intake calculated from the dietary survey was found to be higher in females than in males. When compared by age, the average was lower in the young group and higher in the old group for both males and females. The serum concentrations of carotenoids behaved similarly to the vegetable intake. In terms of differences, the level of zeaxanthin did not differ between males and females, whereas the level of lycopene was higher in younger than in older subjects.”

Also, make sure that your text corresponds to the results in Table 1. For example, in page 4, line 141 it is mentioned that BMI was higher in middle-aged males compared to other age groups but the table indicates that this is the case only compared to the younger group.

Response: Thank you for your comment. The following revision has been made.

(Page 4, Lines 141-142 in the revised version)

“However, two markers (BMI and TG) showed the highest value in middle-aged males and two markers (blood insulin and HDL) began to worsen significantly in older females.”

    1. When referring to regression coefficients please prefer words like association or relationship rather than correlation which mostly refers to correlation coefficients.

Response: Thank you for your comment. As pointed out, we have dropped the word “correlation” in the text and changed it all to “association” or “relationship.” The following revisions were made.

Page 1, Line 26, 27 and 28

Page 7, Line 160, 171 and 180

Page 10, Line 219, 221, 222, 223, 224 and 225

Page 16, Line 294, 305, 307

Page 17, Line 329

Page 18, Line 386

Footnotes of Tables 2, 3, 4, and 5

    1. In Section 3.4.1, please provide a sentence on what are the associations between total carotenoids and CVD markers before comparing them with other exposures like b-carotene and b-cryptoxanthin.

Response: Thank you for your comment. Total carotenoids have been mentioned in the previous chapter, and I have added that information. In addition, to clarify that total carotenoids are a restatement, we added the word “repeated” next to “Total carotenoid” in Tables 4 and 5.

(Page 10, Lines 209-212 in the revised version)

“In the previous chapter, we discussed total carotenoids. As mentioned in the Introduction, different carotenoids may behave differently. Therefore, we also evaluated individual carotenoids. Multiple regression analysis was performed using CVD markers as objective variables and individual carotenoid concentrations as explanatory variables.”

    1. As text under some subheadings is scarce (e.g. page 14, lines 12 and 15), I’d suggest not using subheadings for every model specification. It might make more sense to focus your results on fully adjusted models.

Response: Thank you for your comment. I removed the subheadings and revised some of the sentences as follows.

(Page 7, Lines 164-168 in the revised version)

“Multiple regression analysis was performed using the nine abovementioned CVD markers as objective variables and the serum concentrations of total carotenoids as explanatory variables for the three age categories and in total for each sex. The standardized regression coefficients for each of these are shown in Tables 2 and 3 and are compared in the presence and absence of adjustment factors.”

    1. Page 7, line 3: did you consider including more confounders in this regression?

Response: Thank you for your comment. No other confounding factors were considered in this study. We would like to consider them in the future.

  1. The writing style of the Discussion also needs to be significantly improved. Here are some specific points:

    1. It is important not to repeat your results in the discussion but use this space to compare with relevant up-to-date literature. For example, in page 17, lines 91-108 are just a repetition of results.

Response: Thank you for your comment. We have reviewed our manuscript and removed and/or revised the parts that were duplicated throughout the Discussion. If there are any other areas of concern, please let me know.

For example, the following revisions were made at the part of the Discussion.

(Page 16, Lines 304-311 in the revised version)

“Without age stratification, in males, seven CVD markers (baPWV, SBP, DBP, HOMA-IR, insulin, TGs, and HDL cholesterol) out of the nine were significantly associated with total carotenoids (Table 2), whereas in females, eight markers (BMI, baPWV, SBP, HOMA-IR, blood insulin, FBG, TGs, and HDL cholesterol) were associated with total carotenoids (Table 3). Many studies have shown that some CVD markers are healthier in those with higher vegetable intake [4, 30, 31]. In addition, it has been suggested that the relationship between vegetable intake and some CVD markers is due to antioxidants such as carotenoids in the blood [32, 33]. These results are consistent with ours.”

    1. In sections 4.3 and 4.4 you don’t need to discuss every model specification separately.

Response: Thank you for your comment. We reviewed the entire Discussion and stopped using subheadings by model.

(Page 17, Lines 345-353 in the revised version)

“In addition to dietary habits, lifestyle habits, such as smoking, alcohol intake, and exercise, have a major impact on the development and progression of CVDs [46]. Therefore, these three items were added to the adjustment factors in the analysis. However, no significant change was observed in the results, suggesting that the relationships observed in this study are independent of the lifestyles mentioned above. Obesity also underlies many CVDs [47]. Therefore, BMI was added to the adjustment factors, but no significant change in the relationship between total carotenoids and CVD markers was observed in males or females, suggesting that these relationships are also independent of BMI. Thus, the associations that we found were probably due to the role of carotenoids and other ingredients of vegetables mentioned above.”

(Page 18-19, Lines 413-418 in the revised version)

“The significant relationships between the serum concentration of β-cryptoxanthin and blood pressure or diabetes-related markers in females disappeared when we adopted BMI as the adjustment factor. This might suggest that obesity is a confounder for these associations. This is supported by previous reports, which showed that β-cryptoxanthin acts as an antagonist of peroxisome proliferation-activated receptor gamma (PPARγ) and may improve lipid metabolism and inhibit obesity via the inhibition of adipocyte hypertrophy [58, 59].”

    1. Section 4.4.1 provides a large description of potential mechanisms of identified relationships. This would be clearer to the reader if it was provided in a separate section with a relevant subheading.

Response: Thank you for your comment. Overall, we reviewed the chapter structure of the Discussion. We divided the Discussion in some paragraphs in an easily understood manner and changed the order of the sentences. We also added some sentences as follows.

(Page 16-17, Lines 312–322 in the revised version)

“The best studied physiological function of carotenoids is quenching the activity of singlet oxygen, one of the reactive oxygen species (ROS) [34]. ROS have been considered to elicit chronic inflammation and accelerate the onset and/or progression of CVDs [35]. In the present study, we did not evaluate any circulating biomarkers of early endothelial dysfunction, which is the starting point of the atherosclerotic process and accompanies the progression of CVDs [36]. We suggest that tomatoes can help prevent atherosclerosis by inhibiting vascular endothelial dysfunction in a primitive animal experience [37]. Tsitsimpikou et al. suggested that continuous intake of tomato juice significantly improves the inflammation status and endothelial dysfunction [38]. In the future, we aim to elucidate the mechanism in detail by examining the relationship between internal carotenoid levels and markers such as E-selectin, ICAM-1, VCAM-1, and von Willebrand Factor as well as inflammatory markers.”

    1. In Section 4.2, the authors refer to limitations of food frequency questionnaires used in the study by Deborah et al. (please use author last name here) but it is not clear how the dietary assessment method used in their analysis was superior. Also, there are more recent studies investigating associations between plasma carotenoid and vegetable intake, including systematic reviews e.g. https://doi.org/10.1017/S0007114515003165

Response: Thank you for your comment. In the study by Campbell (Deborah’s last name), as in the present study, vegetable intake was calculated using the frequency questionnaire, and it was correlated with the total amount of multiple carotenoids. In the paper you introduced, a relationship between individual carotenoids and vegetable and fruit intake was shown. Therefore, we cited both references and made the following revisions.

(Page 16, Lines 295-302 in the revised version)

“In a recent systematic review, it was suggested that the concentrations of some carotenoids (lutein, β-cryptoxanthin, α-carotene, and β-carotene) are higher in the high-intake group of fruits and vegetables than in the low-intake group [28]. A relationship between blood carotenoid levels and vegetable intake was also reported by Campbell et al. using a frequency survey called Food Frequency Questionnaire [16]. These previous studies are consistent with ours, suggesting that serum carotenoid concentrations could be an indicator of vegetable intake and that it could be expected to be helpful for nutritional guidance on which non-invasive measurement of skin carotenoid levels has begun to be implemented [29].”

    1. Surprised not to see any comments on the external validity of the study and the potential problems with multiple testing in Section 4.5

Response: Thank you for your comment. As you have pointed out, this study used health examinations in a region of Japan. Although the study population is not considered unique among Japanese individuals, given the results of the National Health and Nutrition Examination Survey, further testing is needed to examine whether it is replicable in populations with different lifestyles and races.

The present study also analyzed nine markers in combination with carotenoids. Hence, we cannot rule out the possibility that we might be looking at an error due to multiple comparisons. However, the present study discusses the differences that have been found to be trending across multiple indicators. Therefore, we do not believe that all the conclusions reached can be rejected.

Therefore, the following revisions were made.

(Page 19, Lines 426-428 in the revised version)

“As this study was performed via a resident-based health examination, the participants were all Japanese living in the narrow area. Therefore, reproducibility should be confirmed in a different country and/or race.”

(Page 19, Lines 442-445 in the revised version)

“We analyzed many CVD markers in combination with carotenoids. It cannot be denied that the significant relationships observed in this study were errors due to multiple comparisons. However, to avoid picking up such errors, we focused on what changed in the series, rather than a single result.”

  1. Methods
    1. It is not clear to me why authors proceeded with age stratification, especially given the potential lack of power in these groups (sample size 151-292). Was there strong indication of effect modification with age? I understand that the authors imply that age-stratification would indicate when carotenoids need to be consumed to prevent CVD but I doubt that a cross-sectional analysis can answer this question.

Response: Thank you for your comment and thank you for understanding our intent. As a result of age group stratification, the number of subjects included in the analysis was reduced. This study was based on a cross-sectional analysis. Therefore, the quality of the evidence in this study was low. However, we believe that the results of this study will be important in designing future causal studies (e.g., longitudinal studies and intervention trials). However, this sense of purpose has not been properly presented in the Introduction, and some parts of it have been overstated in the Discussion. Therefore, the following revisions were made.

(Page 2, Lines 65-73 in the revised version)

“As mentioned above, many lines of evidence have suggested that carotenoids are effective in preventing CVDs. However, a large percentage of healthy people are unaware of the importance of prevention, and it may take some triggers for them to change behaviors like proactive vegetable intake. One of the effective rationales is to provide personalized information. For example, information on when to start eating carotenoid-rich vegetables would be useful. However, to the best of our knowledge, no research has so far provided such information. Therefore, as a first step to obtain such information, we conducted a cross-sectional study on the associations between serum concentrations of carotenoids and various health parameters associated with CVDs, including lifestyle, in a relatively healthy population with different ages.”

(Page 18, Lines 369-371 in the revised version)

“Our findings, obtained by stratifying sex and age, should act as useful references of a new intervention or prospective study to confirm the causal correlation that indicates when we should consume more carotenoid-rich vegetables.”

(Page 19, Lines 432-435 in the revised version)

“As mentioned in the Introduction, we conducted a stratified analysis by age despite the limitation of decreasing power of detection to obtain information on when we should eat carotenoid-rich vegetables to prevent CVDs. Increasing the number of subjects is expected to increase the detection power in the future.”

    1. Please give more details on the BDHQ used, including the time frame of consumption it corresponds, if it was quantitative, created or adapted for local diets, etc.

Response: Thank you for your comment. The BDHQ asks the subjects to fill out a reminder form about the past month’s meals and is a survey in which daily intake values are calculated for major food groups. This is a commonly used and validated dietary survey in Japan. The following revision was made.

(Page 3, Lines 94-96 in the revised version)

“BDHQ calculates the daily intake for key food groups on the basis of the past month’s dietary surveys. Notably, this questionnaire has been validated and is commonly used as a dietary survey in Japan [20].”

    1. Also, did the BDHQ provide information on dietary intake of other groups, such as fruit? Did you consider adjusting for other dietary confounders? Please specify the measurement units of vegetable and alcohol intake.

Response: Thank you for your comment. It is true that fruits are also considered as a source of carotenoids and that fruit intake can be calculated using the BDHQ. However, the main fruits that are considered a source of carotenoids for Japanese people are mandarin oranges, persimmons, and watermelon. As this study was performed at a time different from the season of those fruits, we did not use them as an adjustment factor.

The measuring units used were g/day for both vegetables and alcohol. The following revision was made.

(Page 3, Lines 92-94 in the revised version)

“The volume of alcohol consumed (g/day) and that of vegetables consumed (g/day) were estimated using a brief-type self-administered diet history questionnaire (BDHQ) [19].”

    1. Were height and weight self-reported or measured? If measured, how?

Response: Thank you for your comment. Both weight and height were measured by body measurements. The following revision was made.

(Page 3, Lines 98-99 in the revised version)

“Body mass index (BMI) was calculated from the body height and weight, which were obtained by anthropometric measurements.”

    1. Does smoking habits refer only to current smoking and to all tobacco products?

Response: Thank you for your comment. We confirmed the current smoking status by self-assessment. Therefore, while we are aware that all tobacco products have been included, we cannot rule out the possibility that some participants may have excluded some tobacco products. The following revision was made.

(Page 3, Lines 89-90 in the revised version)

“The questionnaire contained information about sex, age, current smoking habits, exercise habits, medical history, and use of medications.”

    1. page 3, line 127-128s: Please give justification for why you adjusted for serum/plasma vitamins.

Response: Thank you for your comment. Vitamins E and C are antioxidants as well as carotenoids, and some carotenoids, such as β-carotene, have a provitamin A activity. Therefore, we used them as adjustment factors because we thought that they would affect the results. The following revision was made.

(Page 7, Lines 156-159 in the revised version)

“Multiple regression analysis was performed using vegetable intake as an objective variable, blood total carotenoid concentration as an explanatory variable, and antioxidative vitamins (vitamin C [ascorbic acid], vitamin E [⍺-tocopherol], and vitamin A [retinol]), to which some carotenoids were internally converted, as adjustment factors.”

    1. Page 3, line 116: are the results of this correlation analysis presented anywhere?

Response: Thank you for your comment. The results were described in the section 3.4 (Page 10, Line 212-214) as the correlation between α-carotene and β-carotene. Because of the large amount of data, no other carotenoid correlations have been included in this paper. We will submit them as Supplemental Data.

    1. Page 3, line 118: please mention the covariates you used in this regression.

Response: Thank you for your comment. Vitamins A, C, and E have been added to the analysis and the following revision was made.

(Page 4, Lines 122-124 in the revised version)

“Serum concentrations of vitamins A and E and plasma concentration of vitamin C were also adopted as adjustment factors for the evaluation of total carotenoids.”

  1. Introduction:
    1. I am not fully aware of all the relevant literature but it seems that authors tend to cite older papers. Isn’t there any more recent and robust work to be cited? See for example, this seminal meta-analysis by Aune https://doi.org/10.1093/ije/dyw319

Response: Thank you for your comment. A similar point has been raised by another reviewer as well. The following revision was made.

(Page 2, Lines 38-39 in the revised version)

“In a systematic review, Aune et al. showed that the summary relative risk per 200 g of vegetables per day is 0.9 for CVDs [7].”

    1. Please clearly provide the aim of this study in the last paragraph of the introduction.

Response: Thank you for your comment. To clarify the aim of this study, we made the following revisions.

(Page 2, Lines 65-73 in the revised version)

“As mentioned above, many lines of evidence have suggested that carotenoids are effective in preventing CVDs. However, a large percentage of healthy people are unaware of the importance of prevention, and it may take some triggers for them to change behaviors like proactive vegetable intake. One of the effective methods is to provide personalized information. For example, information on when to start eating carotenoid-rich vegetables would be useful. However, to the best of our knowledge, no research has so far provided such information. Therefore, as a first step to obtain such information, we conducted a cross-sectional study on the associations between serum concentrations of carotenoids and various health parameters associated with CVDs, including lifestyle, in a relatively healthy population with different ages.”

  1. Conclusions:
    1. Page 19, lines 202-203: You imply that your study showed associations between vegetable intake and CVD markers which is not the case. Please make sure that the conclusion reflects your results.

Response: Thank you for your comment. A similar point has been raised by another reviewer as well. The following revision was made.

(Page 19, Lines 448-449 in the revised version)

“Higher concentration of serum carotenoids in relatively healthy individuals is associated with better CVD markers.”

    1. Page 19, lines 205-207: It is not clear which associations you are referring to, I assume this is about associations between total carotenoids and CVD markers. It is a bit misleading to focus on these age groups as associations don’t seem to be stronger in these groups compared to others, e.g. middle-aged males and young females.

Response: Thank you for your comment. Our intended meaning was that the number of CVD markers associated with carotenoids in young males and middle-aged females is much higher than in other age groups. As you have pointed out, the sentence was not suitable. Therefore, the following revision was made.

(Page 17, Lines 359-361 in the revised version)

“In analyses with additional adjusted factors (lifestyle, BMI), a prominent number of significant relationships were observed between the serum total carotenoid levels and CVD markers in young males and middle-aged females.”

(Page 19, Lines 451-454 in the revised version)

“In addition, there were more significantly suppressive associations between carotenoids and CVD markers in young males and middle-aged females, respectively, than in other age groups, in which the markers of CVDs begin to deteriorate for both sexes.”

Response: Again, we sincerely appreciate for your constructive comments on our paper. We believe that we have appropriately responded to all of the above comments. We also made some changes according to the other reviewers’ suggestions. All of changes which we made were able to be checked by the tracked changes of the revised manuscript.

Round 2

Reviewer 2 Report

The Authors appropriately corrected the manuscript following the reviewers' suggestions.

In rereading the Discussion, the reviewer hypothesized that the following statement "In the future, we aim to elucidate the mechanism in detail by examining the relationship between internal carotenoid levels and markers such as E-selectin, ICAM-1, VCAM-1, and von Willebrand Factor as well as inflammatory markers." could be reinforced by a recent study published by Ucci and colleagues (Ucci M et al, Anti-inflammatory Role of Carotenoids in Endothelial Cells Derived from Umbilical Cord of Women Affected by Gestational Diabetes Mellitus. Oxid Med Cell Longev. 2019)

Author Response

Response to Reviewer 2 Comments

Comments from Reviewer 2;

The Authors appropriately corrected the manuscript following the reviewers' suggestions.

In rereading the Discussion, the reviewer hypothesized that the following statement "In the future, we aim to elucidate the mechanism in detail by examining the relationship between internal carotenoid levels and markers such as E-selectin, ICAM-1, VCAM-1, and von Willebrand Factor as well as inflammatory markers." could be reinforced by a recent study published by Ucci and colleagues (Ucci M et al, Anti-inflammatory Role of Carotenoids in Endothelial Cells Derived from Umbilical Cord of Women Affected by Gestational Diabetes Mellitus. Oxid Med Cell Longev. 2019)

Response: Thank you for your comment. As you pointed out, the study by Ucci et al. is a very useful study for future investigation of the mechanism of the vascular endothelial protection of carotenoids. Therefore, we have added the following sentence in Discussion.

(Page 16-17, Lines 319-321 in the revised version)

“Regarding these mechanisms, reduction of the total expression of VCAM-1 and ICAM-1 by carotenoids (β-carotene and lycopene) has been reported to act an important role by Ucchi et al [39].”

We also corrected the bibliographic items of some of reference articles.

Thank you again for your kind suggestion.
